# Emerging Therapies in Kirsten Rat Sarcoma Virus (+) Non-Small-Cell Lung Cancer

**DOI:** 10.3390/cancers16081447

**Published:** 2024-04-09

**Authors:** Anastasia Karachaliou, Elias Kotteas, Oraianthi Fiste, Konstantinos Syrigos

**Affiliations:** Oncology Unit, Third Department of Internal Medicine and Laboratory, Medical School, National and Kapodistrian University of Athens, “Sotiria” General Hospital, 11527 Athens, Greece; ilkotteas@med.uoa.gr (E.K.); ofiste@med.uoa.gr (O.F.); ksyrigos@med.uoa.gr (K.S.)

**Keywords:** KRAS, NSCLC, G12C mutation, undruggable, sotorasib, adagrasib, mechanisms of resistance, emerging therapies

## Abstract

**Simple Summary:**

Cancer is a heterogeneous disease characterized by uncontrollable abnormal cell growth. KRAS plays a crucial role in cell proliferation and differentiation. KRAS mutations are frequently found in many types of cancer, including non-small-cell lung cancer. However, targeting this oncogene has been a real challenge for many years. Recent advances in KRAS targeting have changed the current therapeutic landscape. In this review, we summarize the therapeutic strategies for KRAS-mutated NSCLC with encouraging results and restrictions. Moreover, we present possible combination therapies aiming to achieve individualized treatment.

**Abstract:**

Kirsten rat sarcoma virus (KRAS) is the most frequently found oncogene in human cancers, including non-small-cell lung cancer (NSCLC). For many years, KRAS was considered “undruggable” due to its structure and difficult targeting. However, the discovery of the switch II region in the KRAS-G12C-mutated protein has changed the therapeutic landscape with the design and development of novel direct KRAS-G12C inhibitors. Sotorasib and adagrasib are FDA-approved targeted agents for pre-treated patients with KRAS-G12C-mutated NSCLC. Despite promising results, the efficacy of these novel inhibitors is limited by mechanisms of resistance. Ongoing studies are evaluating combination strategies for overcoming resistance. In this review, we summarize the biology of the KRAS protein and the characteristics of KRAS mutations. We then present current and emerging therapeutic approaches for targeting KRAS mutation subtypes intending to provide individualized treatment for lung cancer harboring this challenging driver mutation.

## 1. Introduction

For several decades, lung cancer has been the deadliest form of cancer worldwide. The majority of lung cancer cases (approximately 85%) belong to the non-small-cell lung cancer type, especially the adenocarcinoma histological subtype [1,2]. In recent years, the introduction of targeted therapies with tyrosine kinase inhibitors (TKIs) has significantly changed the therapeutic management of NSCLC [3,4]. Molecular characterization of the tumor at baseline gives us the opportunity to identify those patients with oncogenic alterations and offer them the optimal therapeutic strategy, thus improving their survival and quality of life [5,6]. Among all the genetic alterations found in NSCLC, Kirsten rat sarcoma virus (KRAS) is the most common oncogene that serves as a driver mutation in approximately 30% of all cases [7,8]. For many years, the KRAS protein was considered an “undruggable” target due to the failure of many therapeutic approaches [9]. However, recently, promising KRAS inhibitors were designed, leading to a new era of effective KRAS-targeting agents [10]. In this review, we provide an overview of the structure and function of the KRAS protein and discuss the emerging therapies targeting KRAS, shedding light on future perspectives and restrictions.

## 2. KRAS Biology

### 2.1. Structure and Signaling Pathways

The Kirsten rat sarcoma virus (KRAS) gene was first discovered in 1982 and belongs to the RAS gene family along with the Harvey rat sarcoma virus (HRAS) and the neuroblastoma rat sarcoma virus (NRAS) [11]. It is located in the short arm of chromosome 12 (12p), and it encodes two alternative spliced variants: KRAS 4A and KRAS 4B [12,13]. 

Structurally, KRAS protein is composed of two main regions: the catalytic domain (G-domain) and the C-terminal hypervariable region (HVR). The G-domain consists of the P-loop and switch regions (I and II) and is responsible for the interaction and binding with regulatory proteins. It enables the GTP function of the KRAS protein. HVR is required to bind the protein to the cell membrane, a necessary step for it to be activated [14,15,16]. 

Under normal conditions, KRAS functions as a GTPase protein that relays extracellular signals to the nucleus, leading to the activation of several intracellular pathways that regulate proliferation, differentiation, survival, and apoptosis [17,18]. GTP and GDP molecules are required for the activation and inactivation of KRAS. It acts like a molecular switch that is “turned on” (activated) when bound to GTP and “turned off” (inactivated) when bound to GDP [17,19]. The alteration between the two function states of KRAS is mediated by regulatory proteins. Guanine nucleotide-exchange factors (GEFs), including Son of Sevenless (SOS 1), promote the GTP-bound form of KRAS by the release of GDP and GTPase-activating proteins (GAPs), such as neurofibromin 1 (NF1), hydrolyze GTP to GDP, thus deactivating the protein [20,21,22]. In basal conditions, KRAS remains bound to the GDP in an inactivated form. When extracellular stimuli bind and activate the receptor tyrosine kinase (RTK) on the cell membrane, SOS 1 promotes the substitution of GDP to GTP with a conformational change in the switch I and II regions [10,13,23]. The activation of KRAS is also mediated by the SHP2 protein, which interacts with the RTK and SOS 1 [24,25]. The GTP-bound KRAS stimulates multiple downstream pathways, including the mitogen-activated protein kinase (MARK) pathway (RAF–MEK–ERK), the phosphatidylinositol 3-kinase (PI3K)–AKT–mammalian target of rapamycin (mTOR) pathway and the RAS-like (RAL) pathway [14]. When a mutation in KRAS occurs, it becomes an oncogene. GAP is hindered from binding to KRAS, blocking GTP hydrolysis and maintaining the protein in an active state. The constitutive stimulation of the downstream signaling cascades leads to oncogenic cell proliferation and tumorigenesis [26,27]. 

### 2.2. Cooperation of KRAS and WNT Pathways 

According to studies, KRAS and WNT/β-catenin pathways share a significant connection leading to cancer development and progression, especially colorectal cancer (CRC). Gene mutations such as APC and KRAS mutations are frequently detected in CRC, and their interactions promote tumorigenesis. The loss of APC causes the stabilization of β-catenin and KRAS. The cooperation of WNT/β-catenin and KRAS-ERK pathways is associated with tumor growth and treatment resistance. Consequently, this intriguing crosstalk stands as a potential therapeutic avenue for cancer treatment [28,29,30,31]. 

### 2.3. Mutations

KRAS is the most common driver mutation in NSCLC, discovered in approximately 30% of all cases [11]. The majority of alterations are located in codon 12 (90%), followed by codons 13 and 61. The KRAS-G12C mutation, characterized as a replacement of glycine with cysteine at codon 12, is the prevailing KRAS mutation (40% of KRAS mutations) found in 13% of all NSCLC cases [8,32]. Substitution of glycine with valine (KRAS G12V) and substitution of glycine with aspartic acid (KRAS G12D) are less frequent alterations [33]. KRAS mutations have been associated with characteristics such as the female sex, adenocarcinoma histology, and smoking habits [34,35]. However, this depends on the specific KRAS mutation subtype. So, among smokers, it is more likely to discover the G12C or G12V mutations, whereas G12D is usually found in patients who do not smoke [32]. Moreover, the incidence of KRAS alterations is higher in Western countries compared with Asian countries [36,37]. Although the prognostic role of KRAS mutations remains a matter of debate, some studies have associated patients harboring these mutations with lower progression-free survival (PFS) and overall survival (OS) [38,39]. However, these survival outcomes need to be confirmed in clinical trials.

### 2.4. Comutations

KRAS mutations may be mutually exclusive with oncogenic drivers such as EGFR and ALK, but they often coexist with mutations in other genes [40,41]. The most common concurrent molecular alterations are TP53, STK11, and KEAP1. KRAS and its comutations are categorized into three clusters: (1) KRAS with TP53 mutation (KP group), (2) KRAS with STK11/LKB1 mutation including the KEAP1 mutation (KL group), and (3) KRAS with inactivation of CDKN2A/B (KC group) [42]. According to studies, STK11 and KEAP1 mutations are negative prognostic factors and forecast impaired OS [43]. Furthermore, some comutations have been associated with an altered immunogenic profile. Skoulidis et al. elucidated the lower expression of PD-L1 and deficient tumor microenvironment when the STK11 comutation is present and the higher levels of tumor-infiltrating lymphocytes (TILs) and PD-L1 when the TP53 mutation coexists [42,44,45]. All the above data propose the key role of concomitant mutations on the heterogeneity of KRAS-mutated tumors and their impact on different therapeutic strategies [46].

### 2.5. Why Was the KRAS Mutation Undruggable?

For many years, all efforts of therapeutic targeting KRAS mutations had been unsuccessful, leading to its characterization as an “undruggable” target. The main reasons making this process difficult were the structure and complexity of KRAS pathways [47,48]. KRAS protein consists of a highly smooth surface and a unique GTP binding site. The absence of available binding pockets, along with the protein’s high affinity for GTP, have made the development of effective targeted agents a challenge [49]. However, the discovery of the switch II pocket on the KRAS-G12C-mutated protein by Shokat lab in 2013 paved the way for the design of agents that selectively bind to the cysteine residues [50]. Even though many molecules have been developed since then, only two of them, sotorasib (AMG510) and adagrasib (MRTX849), have demonstrated clinical efficacy and received FDA approval [10,13].

## 3. Direct Targeting of KRAS

### 3.1. Sotorasib (AMG510)

Sotorasib is the first clinically effective selective inhibitor of KRAS-G12C mutation that gained FDA approval [51]. Lumakras (Sotorasib) is an oral small molecule that covalently binds to the cysteine in the switch II pocket, blocking the interaction between KRAS and SOS 1 and locking KRAS in its inactive GDP-bound state [50]. CodeBreak 100 was a phase I/II study that evaluated the safety, pharmacokinetics, and efficacy of sotorasib in patients with locally advanced or metastatic KRAS-G12C tumors who had already received previous lines of therapy [52,53]. In the phase I trial, safety was the primary endpoint. After 11.7 months of median follow-up, the median PFS was 6.3 months. The dose of 960 mg daily was set as the dose for the expansion cohort. The most frequently reported treatment-related adverse events (TRAEs) were diarrhea, fatigue, nausea, vomiting, and abnormal levels of aminotransferases [52]. In phase II of the CodeBreak 100 trial, the patients enrolled had progressed after platinum-based chemotherapy and immunotherapy alone or combined. After 15.3 months of follow-up, the median PFS and OS were identified as 6.8 months and 12.5 months, respectively. The TRAEs reported were similar to the ones in the phase I trial and included gastrointestinal AEs and fatigue. In an exploratory analysis of the phase II trial, Skoulidis et al. investigated the relation between comutation subgroups and the efficacy of sotorasib treatment. Patients with STK11 mutations and wild-type KEAP1 demonstrated better response (50%) in comparison with mutated KEAP1 and wild-type STK11 (14%) [53]. Moreover, according to a post hoc analysis, Lumakras showed intracranial efficacy in brain metastases [54,55]. Based on the promising results of CodeBreak 100, sotorasib received accelerated approval from the FDA in May 2021 for patients with KRAS-G12C-mutated NSCLC who had been treated with at least one prior line of anticancer therapy [56]. 

Additional studies were designed to evaluate sotorasib as monotherapy or in combination with other targeted agents, immunotherapy, and chemotherapy. CodeBreak 200 is a global, randomized, open-label phase III trial aiming to assess the activity of sotorasib versus docetaxel in the second line for patients with advanced KRAS-mutated NSCLC. Preliminary data were published at ESMO 2022. The KRAS-G12C inhibitor was associated with a better PFS and safety profile compared to docetaxel but showed no difference in terms of OS [57,58,59].

In another phase II study, known as CodeBreak 201 (NCT04933695), sotorasib is being evaluated as a first-line therapy in patients with KRAS-mutant NSCLC with the following criteria: (1). stage IV, (2). PD-L1 TPS score < 1%, and/or (3). comutated STK11 [52]. Finally, CodeBreak 101 (NCT041185883) is an ongoing phase Ib/II study testing the efficacy and safety of sotorasib in combination with targeted molecules, such as EGFR, SHP2, mTOR, and CDK inhibitors, but also with chemotherapy and immune checkpoint inhibitors (ICIs) in advanced KRAS-G12C-mutated lung cancer [60].

### 3.2. Adagrasib (MRTX849)

Adagrasib is the second direct KRAS-G12C inhibitor that demonstrated sufficient clinical activity. The mechanism of action is similar to sotorasib and is based on the irreversible bind to the protein, trapping it in its inactive GDP-bound state. Interestingly, adagrasib has shown superiority in terms of pharmacokinetic properties such as long half-life (approximately 24 h), elevated oral bioavailability, and extensive tissue distribution. In addition, adagrasib remains efficient even at low concentrations and is highly selective for the KRAS-G12C mutation [61]. 

KRYSTAL-1 was the phase I/Ib study that evaluated the safety and efficacy of adagrasib in patients with KRAS-G12C-mutated NSCLC [62,63,64]. After 19.6 months of follow-up, the median PFS was 11.1 months. The recommended dose for phase II (RP2D) was identified as 600 mg twice daily. The most commonly experienced treatment-related adverse (TRAEs) were nausea, diarrhea, vomiting, and fatigue [62]. In the phase II expansion cohort of KRYSTAL-1, the patients enrolled were already treated with chemotherapy and/or immunotherapy in previous lines. They received 600 mg of adagrasib per os twice a day, and after follow-up, the reported mPFS and mOS were 6.5 months and 12.6 months, respectively. The safety profile included gastrointestinal AEs (diarrhea, nausea, vomiting), fatigue, and increased levels of aminotransferases (ALT and AST). Moreover, in patients with stable, previously treated brain metastases, the intracranial confirmed response rate (ORR) was 33.3%. As for comutations, KEAP1 was associated with worse responses [65,66]. Based on the aforementioned results, adagrasib obtained FDA approval for patients with previously treated KRAS-G12C mutation [67]. 

Ongoing clinical trials aim to investigate novel treatment approaches with the combination of adagrasib with other anticancer drugs. The KRYSTAL-12 (NCT04685135) phase III trial tests the clinical activity of adagrasib versus docetaxel in pretreated patients [68,69]. Another phase II/III study (KRYSTAL-7) uses adagrasib with pembrolizumab in the first line and evaluates the efficacy in three subgroups: 1. PD-L1 TPS < 1%—combination, 2. PD-L1 TPS < 1%—adagrasib monotherapy, and 3. PD-L1 TPS > 1%—combination (NCT04613596) [70]. Finally, the associations between the KRAS inhibitor and other agents, including BI 1701963 (SOS inhibitor), TNO155 (SHP2 inhibitor), and palbociclib (CDK4/6 inhibitor), are being investigated in the studies KRYSTAL-14 (NCT04975256), KRYSTAL-2 (NCT04330664), and KRYSTAL-16 (NCT05178888), respectively. 

### 3.3. Novel Direct Inhibitors of KRAS G12C

Many phase I/II clinical trials are ongoing for the evaluation of newly designed KRAS-G12C inhibitors. 

GDC-6036 is a novel drug that is being tested in a phase I trial (NCT04449874) as monotherapy or in combination with other molecules, including bevacizumab, atezolizumab, and cetuximab in patients with KRAS-mutated tumors. According to the preliminary results presented at the 2022 World Congress on Lung Cancer (WCLC), the ORR was 46%. As for treatment-related adverse events (TRAEs), the most commonly reported were nausea, diarrhea, vomiting, fatigue, elevated liver enzymes (ALT and AST), and reduced appetite [71,72]. 

Another novel molecule that selectively inhibits KRAS-G12C mutation is D-1553. In the phase I study (NCT04585035), where pretreated patients were enrolled, it demonstrated an ORR of 40.4% [73]. 

Moreover, JDQ443 is currently being tested in a phase I/II trial (NCT04699188) as monotherapy and in combination with TNO155 (SHP2 inhibitor) or tislelizumab (antiPD1 antibody). Finally, JAB-21822, LY3537982 (NCT04956640), RMC6291 (NCT05462717), and multiple novel agents have entered clinical trials and are currently under investigation (Table 1). 

## 4. Resistance to KRAS-G12C Inhibitors

Despite the aforementioned encouraging clinical outcomes that highlight the efficacy of novel KRAS-G12C inhibitors, the possibility of patients demonstrating disease progression is high. This can be attributed to a variety of resistance mechanisms that are divided into two types: intrinsic and acquired mechanisms of resistance. 

Regarding intrinsic resistance, one of the main reasons for the limited response to KRAS inhibitors is the low dependence on KRAS signaling. The RAS protein regulates multiple downstream pathways that, in some cases, can be activated even after KRAS knockdown [74,75]. Specifically, the PI3K pathway can be stimulated even in the presence of KRAS inhibitors [76]. Moreover, secondary KRAS mutations at baseline result in a heterogeneous response to direct inhibitors of KRAS-G12C mutation [77]. 

As for the acquired type of resistance, two mechanisms have been identified: on-target and off-target [78]. The on-target mutations impact the switch II region, which is essential for successful drug binding. KRAS G12D/R/V and other secondary mutations have been reported in accordance with the inhibitor that was used for treatment. KRAS G13D has been associated with sotorasib-induced resistance, and KRAS Q99L with adagrasib-induced resistance. Y96D has been observed after exposure to either one of them [79,80]. RM-018 is a novel inhibitor of the KRAS GTP-bound (active) form, effective in overcoming the secondary KRAS-G12C/Y96D mutations [81]. Furthermore, resistance to KRAS-G12C inhibition can arise due to off-target mechanisms. Multiple bypass signaling pathways have been identified to date, including MET amplification; mutations activating NRAS, BRAF, MAP2K1, and RET; fusions of ALK, RET, BRAF, RAF1, and FGFR3; and loss-of-function of PTEN and NF1 [82]. Another way for acquiring resistance is epithelial-to-mesenchymal transformation (EMT) induced by the activation of the PI3K pathway [40]. In some cases, a histological transformation of lung adenocarcinoma to squamous cell carcinoma can be observed [82]. In addition, senescence resulting from the activation of aurora kinase signaling can induce resistance [10]. 

Moreover, the inhibition of KRAS and, subsequently, the block of the MAPK pathway can cause the synthesis of a novel KRAS-G12C protein, which remains active by the receptor tyrosine kinase (RTK), even if an inhibitor is present [83]. 

Finally, the activation of wild-type RAS (HRAS and NRAS) mediated by RTKs is supposed to be another mechanism of resistance to direct KRAS inhibitors (Figure 1) [75,84]. 

In conclusion, an abundance of intrinsic and acquired mechanisms are responsible for the progression of disease in patients harboring KRAS-G12C mutation and treated with direct inhibitors. However, many studies still need to be conducted in order for us to understand those mechanisms fully and develop the optimal treatment strategies to overcome them. For this purpose, efforts to combine KRAS inhibitors with targeted agents depending on the type of resistance are currently being made, and results are eagerly expected [85,86].

## 5. Indirect Inhibition of KRAS

As mentioned before, the development of resistance has overshadowed the encouraging clinical efficacy of novel direct KRAS-G12C inhibitors. The combination of inhibitors with targeted agents of the upstream, downstream, or other important signaling pathways is a promising therapeutic approach and a potential solution to this predicament [85,86]. 

### 5.1. Inhibitors of the Upstream Signaling Pathway

EGFR inhibitors

RTKs, such as EGFR, can induce the rebound activation of KRAS and its downstream pathways as a result of the inhibition of KRAS. EGFR inhibitors, in combination with KRAS-G12C inhibitors, can possibly overcome the resistance mechanism [61]. 

Combinations are already being tested in studies, including CodeBreak 101 (sotorasib with a pan-ErbB inhibitor) and a phase I trial (GDC-6036 with erlotinib—NCT04449874). 

SHP2 and SOS1 inhibitors

Other promising target proteins for the indirect inhibition of KRAS are SHP2 and SOS1. These molecules are essential for the activation of KRAS as part of the upstream pathway [87]. SHP2 inhibitors, including RMC-4630 and TN0155, have been designed and are being evaluated in studies as monotherapy or in combination with KRAS inhibitors (NCT03634982, NCT03114319, NCT04185883). Similarly, BI 1701963 is a pan-KRAS SOS1 inhibitor under investigation in a phase I trial as a single agent or in association with trametinib (MEK inhibitor) (NCT04111458). 

### 5.2. Inhibitors of the Downstream Signaling Pathway 

MAPK pathway (RAF/MEK/ERK)

Regarding the MAPK pathway, the GTP-bound KRAS protein activates RAF, which stimulates the phosphorylation of MEK and then ERK, leading to the activation of transcription factors [88,89,90,91]. RAF inhibitors (belvarafenib) and MEK inhibitors (binimetinib, cobimetinib) are being tested for the indirect inhibition of KRAS (NCT03284502, NCT01859026, NCT01986166). Also, RAF–MEK inhibitor VS-6766 is being used as monotherapy or in combination with defactinib (FAK inhibitor) [92], sotorasib, or adagrasib (KRAS-G12C inhibitors) in ongoing trials (NCT05074810, NCT05375994). The resistance induced by the treatment with RAF or MEK inhibitors led to the development and evaluation of ERK inhibitors (JSI-1187-01) (NCT04418167). The combination of a RAF/MEK inhibitor with a FAK inhibitor and the combinations of MEK and PI3K/mTOR/AKT inhibitors are potent therapeutic strategies targeting the downstream signaling pathway of KRAS [93]. 

PI3K/AKT/mTOR pathway

The PI3K/AKT/mTOR is another well-known downstream pathway activated by KRAS, which contains potential targets for effective inhibition. PI3K inhibitors (BKM120, serabelisib—NCT04073680, XL147), an AKT inhibitor (TAS0612—NCT04586270), and an mTOR inhibitor (A2D2014—NCT02583542) were designed and are being investigated in phase I/II studies [94,95,96]. In addition, the combination of PI3K inhibitors with ARS1620 (second-generation KRAS-G12C inhibitor) has demonstrated efficacy in preclinical studies [97]. 

### 5.3. FAK Inhibitors 

Focal adhesion kinase (FAK) is a non-receptor tyrosine kinase that can cause KRAS mutant cell death if inhibited [98]. The FAK inhibitor defactinib was evaluated in a phase II study (NCT01951690) in previously treated patients showing average clinical activity [99]. 

### 5.4. DDR1 Inhibitors

Discoidin domain receptor 1 is being evaluated as a novel biomarker and therapeutic target in KRAS mutant NSCLC. DDR1 is a receptor tyrosine kinase with a significant role in cell proliferation and differentiation. According to studies, the upregulation of DDR1 in lung cancer is associated with lymph node metastasis, increased invasiveness, and poor prognosis. Moreover, the DDR1–BCR–ABL axis is strongly correlated with the KRAS PI3K–AKT–mTOR axis. The combinational targeting of DDR1, BCR-ABL, and EGFR-ERBB2 is a promising therapeutic approach with future perspectives [28,100,101].

### 5.5. YB-1 Inhibitors

Y-box-binding protein 1 (YB-1) serves as an oncoprotein that plays a crucial role in cancer cell proliferation, stemness, DNA repair, and chemotherapy resistance. YB-1 is activated by the PI3K/AKT and MAPK/ERK pathways, with phosphorylation at S102 mediated by the p90 ribosomal S6 kinase (RSK). RSK inhibitors, such as LJI308, were designed; however, their long-term use leads to AKT reactivation. As a result, the combination of AKT and RSK inhibitors can be a possible solution and an effective therapeutic strategy [102,103]. 

### 5.6. HSP90 Inhibitors 

Heat shock protein 90 (HSP90) is a chaperone responsible for protein stabilization, adjustment to heat, and cell proliferation [104]. Ganetespib, an HSP90 inhibitor, was efficient in blocking tumorigenesis in a phase II study [105]; however, its clinical activity was inadequate in the phase III study (GALAXY II) with docetaxel in KRAS-mutated NSCLC [106]. A novel HSP90 inhibitor, SNX-5422, was associated with promising results in a phase I study with chemotherapy (Table 2) [107]. 

### 5.7. CDK 4/6 Inhibitors 

Cyclin-dependent kinases (CDKs) 4/6 are regulatory agents of the cell cycle, and their activation is connected with the KRAS pathway [108]. Although monotherapy abemaciclib or palbociclib in KRAS-mutant NSCLC was ineffective [109,110], the combination of CDK 4/6 inhibitors with KRAS direct inhibitors seems promising according to preclinical data [74]. Clinical trials are ongoing (CodeBreak 101—sotorasib with palbociclib and KRYSTAL 16—adagrasib with palbociclib—NCT04185883). 

## 6. Platinum-Based Chemotherapy in KRAS-Mutated NSCLC

Despite the tremendous progression made in targeting the KRAS mutation in recent years, platinum-based chemotherapy remains the standard of care for patients with KRAS-mutant NSCLC. But how does the KRAS oncogene influence the response to platinum-based chemotherapy? The data are scarce. However, several studies conclude that KRAS mutation is associated with shorter PFS or OS when patients are treated with traditional chemotherapy. Moreover, among KRAS mutation subtypes, the KRAS G12V is associated with the worst PFS. As a result, KRAS oncogene is a negative predictive factor of PFS in patients who receive first-line platinum-based chemotherapy, but treatment responses differ among KRAS mutation subtypes [111,112]. 

## 7. Immunotherapy in KRAS-Mutated NSCLC

In recent years, the advent of immune checkpoint inhibitors (ICIs) has significantly altered the therapeutic approach of NSCLC [113,114,115,116]. According to preclinical data, patients harboring the KRAS mutation can benefit from monoclonal antibodies targeting PD1 and PD-L1. KRAS-mutated lung cancer has been associated with smoking habits and, consequently, an inflammatory tumor microenvironment (TME) and high tumor mutation burden (TMB) [117,118]. Moreover, CD8+ tumor-infiltrating lymphocytes (TILS) and PD-L1 expression are increased when the KRAS mutation is present [119,120,121,122]. 

The efficacy of immunotherapy in KRAS-mutated NSCLC is also evident in clinical trials. In the CheckMate 057 trial, the KRAS mutation was linked with higher OS when comparing nivolumab (anti-PD-1 antibody) with chemotherapy (HR 0.52, 95% CI: 0.29–0.95) [113]. In addition, in the phase III OAK study, the benefit of atezolizumab was observed in KRAS-mutated patients (mOS: HR = 0.71 [95% CI: 0.38–1.35]) [115]. Similarly, based on the results of the KEYNOTE-042 study (mOS KRASm vs. wt: 21.1 vs. 13.6 months, *p* = 0.03) and the IMMUNOTARGET study (mPFS: 3.2 months and mOS 13.5 months), immunotherapy can be therapeutically exploited in KRAS mutations [123,124]. 

However, it is under consideration whether KRAS point mutations or co-occurring mutations influence the efficacy of ICIs. KRAS-G12D-mutated tumors have been associated with decreased levels of PD-L1, CD8+ TILS, and TMB. These characteristics suggest a lower response to immunotherapy in KRAS-G12D mutation [125,126]. On the contrary, other studies do not verify different results between KRAS mutations [122]. Furthermore, as mentioned before, STK11 or KEAP1 comutations have been linked with decreased PD-L1 and TILS levels, resulting in impaired activity of ICIs [44,127,128]. On the other hand, tumors harboring TP53 comutation benefit more from immunotherapy due to increased expression of PD-L1 [128,129,130]. More studies need to be conducted in order to evaluate the aforementioned outcomes. Among the novel therapeutic approaches targeting the KRAS mutation, the combination of KRAS-G12C inhibitors with immunotherapy seems promising. In preclinical studies, sotorasib induced an inflammatory tumor microenvironment and enhanced the immune system [74]. Similar results were observed in tumors treated with adagrasib [131]. The synergistic effect of KRAS-G12C inhibitors and ICIs is currently being evaluated in phase I/II studies, including the CodeBreak 100 and 101 trials (sotorasib with PD-1/PD-L1 monoclonal antibodies) [132] and the KRYSTAL-7 study (adagrasib with pembrolizumab) [133].

## 8. KRAS-G12V and -G12D Mutations

KRAS-G12V mutation, where glycine is substituted with valine at position 12, is observed in 21% of patients with KRAS-mutated tumors in the lung, especially smokers [134]. This mutation has been associated with impaired OS and increased recurrence rate [135]. Regarding the treatment strategies, chemotherapeutic agents with higher efficacy in KRAS-G12V mutation are taxanes and cisplatin [136,137]. In addition, immunotherapy is another potential approach due to the increased expression of PD-L1 [138]. Emerging therapies with KRAS-G12V inhibitors, including JAB23000, are currently under investigation [87,139]. 

KRAS-G12D mutation, where glycine is replaced with aspartic acid at position 12, has been reported in 15% of KRAS-mutated cases and is usually observed in nonsmokers [134]. As mentioned before, KRAS G12D has been linked with a lower level of PD-L1 protein and immune suppression, resulting in decreased activity of ICIs [140]. As for the direct inhibition of the G12D allele, MRTX1133 is a novel molecule targeting both active and inactive states of the protein and has shown promising activity in preclinical studies. Recently, the US FDA approved the initiation of a phase I study for the evaluation of this first-in-class inhibitor [141]. 

## 9. Other Emerging Therapies

Except for the conventional inhibitors directly targeting the KRAS-G12C protein or molecules of the upstream or downstream KRAS pathways, there is a need for new treatment strategies, including PROTACs, pan-KRAS inhibitors, and agents targeting metabolic rewiring processes and regulatory molecules of the immune system. Proteolysis-targeting chimeras (PROTACs) are emerging targeted agents that degrade the protein of interest using cellular proteasomal degradation mechanisms [142]. KRAS-G12C PROTACs, such as LC-2, have been designed to exploit KRAS inhibitors (MRTX849 in the case of LC-2) for the degradation of KRAS-G12C protein [143]. 

Another novel therapeutic approach targeting multiple KRAS mutations (G12D, G12V, G13D, and G12C) is an mRNA-based cancer vaccine. V941 is being tested in a phase I study (NCT03948763) as a single agent or in combination with pembrolizumab in three different types of cancer, including KRAS-mutated NSCLC [144]. 

Regarding the metabolic rewiring of cancer cells, autophagy plays a crucial role in cell homeostasis. Autophagy inhibition is a potential therapeutic strategy aiming to achieve cancer cell death [145]. Hydroxychloroquine is an autophagy inhibitor that is currently being evaluated in a study along with MEK-inhibitor binimetinib for the treatment of patients with KRAS-mutated lung cancer (NCT04735068). 

Moreover, the KRAS oncogene influences metabolism and especially upregulates the expression of fatty acid synthase (FASN), the enzyme of de novo lipogenesis, in order to prevent lipid peroxidation and, consequently, ferroptosis. Ferroptosis is an oxidative-stress-dependent type of programmed cell death. According to these data, the induction of ferroptosis could be an effective antitumor strategy. TVB-2640 is a FASN inhibitor that is being studied in a phase II trial (NCT03808558) with promising results [146]. 

Furthermore, the intriguing relationship between KRAS and miRNA needs to be explored. Physiologically, miRNAs are responsible for post-transcriptional gene regulation through interaction with target mRNAs. However, KRAS mutations tend to misregulate the miRNA regulatory pathway in multiple ways, including miRNA biosynthesis, expression, and function. As a result, the miRNA machinery can be therapeutically exploited in KRAS-mutated cancers. Studies need to be conducted in order to navigate novel strategies, including simultaneous targeting of KRAS and miRNA pathways [147]. 

Finally, the remodeling of the tumor microenvironment (TME) by cancer cells is another key to the achievement of effective targeted therapy [148]. According to studies, cytokines IL-1β and IL-6 have been associated with cancer cell proliferation due to signal transduction through multiple pathways [149,150,151]. Several trials are ongoing for the evaluation of interleukin inhibitors. Canakinumab is an anti-IL-1β agent under investigation as a monotherapy (NCT03447769) or in combination with immunotherapy (pembrolizumab—NCT03968419 and durvalumab—NCT04905316) or chemotherapy (docetaxel—NCT03626545). Similarly, tecilizumab (IL-6 inhibitor) is being tested in association with ICIs (atezolizumab—NCT04691817 and nivolumab/ipilimumab—NCT04940299) for the treatment of NSCLC, including tumors harboring the KRAS mutation (Table 3 and Table 4). 

## 10. Conclusions

The thought of KRAS as “undruggable” for many years multiplied the efforts for the development of effective targeted therapies. Sotorasib and adagrasib were the first approved direct KRAS-G12C inhibitors, altering the therapeutic landscape of KRAS. However, the emergence of intrinsic or acquired resistance mechanisms limits the duration of activity and efficacy of the novel KRAS-G12C inhibitors. Many ongoing studies are evaluating combination approaches in order to overcome resistance. Moreover, the heterogeneity and complexity of KRAS-mutated tumors are the main reasons why they exhibit different sensitivity to different therapies. According to studies, KRAS mutation subtypes, as well as comutations, have a significant impact on the efficacy of immunotherapy. Molecular profiling of tumors at baseline is an emerging need in order to determine the optimal therapeutic regimen and potential sequencing of therapies. In addition, results of novel strategies, including cancer vaccines, PROTACs, and pan-KRAS inhibitors, are eagerly awaited so as to be introduced in clinical practice. In conclusion, KRAS remains one of the most attractive target mutations in NSCLC, and personalized therapy seems to be the key to the achievement of successful and durable treatment in the future. 

## Figures and Tables

**Figure 1 cancers-16-01447-f001:**
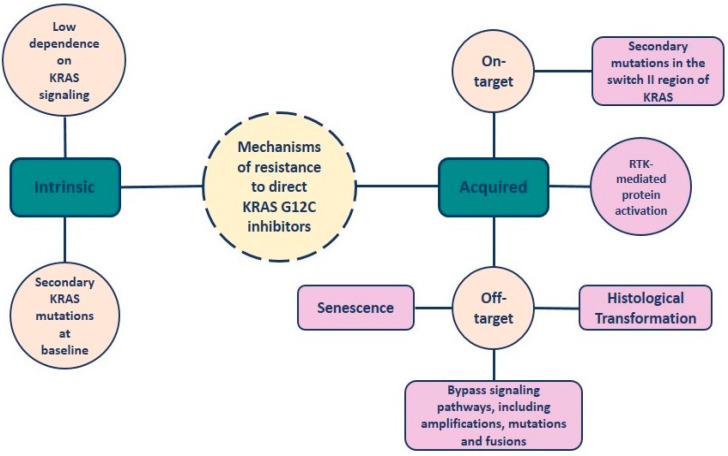
Resistance mechanisms to direct KRAS-G12C inhibition.

**Table 1 cancers-16-01447-t001:** Direct KRAS-G12C inhibitors.

Trial	Phase	KRAS Inhibitor	Combinations	Status/Outcomes	References
CodeBreak 100 NCT03600883	I/II	sotorasib	monotherapy	mPFS 6.8 months mOS 12.5 months FDA Approval	[52,53,56]
CodeBreak 200 NCT04303780	III	sotorasib	monotherapy vs. docetaxel	mPFS 5.6 months vs. 4.5 months (HR 0.66)	[57,58,59]
CodeBreak 201 NCT04933695	II	sotorasib	monotherapy first line (st. IV, TPS <1% and/or comut. STK11)	ongoing	[52]
CodeBreak 101 NCT04185883	Ib/II	sotorasib	EGFR inh, SHP2 inh, mTOR inh, CDK inh. Chemotherapy, ICIs, etc.	Ongoing	[60]
KRYSTAL-1 NCT03785249	I/II	adagrasib	monotherapy/afatinib, pembrolizumab, cetuximab	mPFS 6.5 months mOS 12.6 months FDA Approval	[62,63,64,67]
KRYSTAL-12 NCT04685135	III	adagrasib	monotherapy vs. docetaxel	ongoing	[68,69]
KRYSTAL-7 NCT04613596	II/III	adagrasib	pembrolizumab/pembrolizumab + adagrasib vs. pembrolizumab + chemotherapy (first line)	ongoing	[70]
KRYSTAL-14 NCT04975256	I/Ib	adagrasib	BI 1701963 (SOS inhibitor)	completed	
KRYSTAL-2 NCT04330664	I/II	adagrasib	TNO155 (SHP2 inhibitor)	ongoing	
KRYSTAL-16 NCT05178888	I/Ib	adagrasib	palbociclib (CDK4/6 inh.)	ongoing	
NCT04449874	I	GDC-6036	monotherapy/ bevacizumab, atezolizumab, cetuximab, erlotinib, etc.	ongoing	[71,72]
NCT04585035	I/II	D-1553	monotherapy/other agents	ongoing	[73]
KontRASt-01 NCT04699188	I/II	JDQ443	monotherapy/ TNO155, tislelizumab	ongoing	
NCT05009329	I/II	JAB-21822	monotherapy	ongoing	
NCT04956640	I	LY3537982	monotherapy/ abemaciclib, cetuximab, pembrolizumab, chemotherapy, etc.	ongoing	
NCT05462717	I/Ib	RMC-6291	monotherapy	ongoing	

**Table 2 cancers-16-01447-t002:** Indirect KRAS-G12C inhibitors.

Trial	Phase	KRAS Inhibitor	Combinations	Status/ Outcomes	References
NCT03634982	I	RMC-4630 (SHP2 Inhibitor)	monotherapy	completed	[87]
NCT03114319	I	TNO155 (SHP2 Inhibitor)	monotherapy	ongoing	[87]
NCT04111458	I	BI 1701963 (SOS1 Inhibitor)	monotherapy/ trametinib (MEK Inh.)	ongoing	[87]
NCT03284502	I	belvarafenib (RAF Inh.)	cobimetinib, cetuximab	ongoing	[88,89,90,91]
NCT01859026	I	binimetinib (MEK Inh.)	erlotinib	completed	[88,89,90,91]
NCT01986166	I	cobimetinib (MEK Inh.)	duligotuzumab (EGFR/HER3 Inh.)	completed	[88,89,90,91]
RAMP203 NCT05074810	I/II	Avutometinib [VS-6766] (RAF–MEK Inhibitor)	sotorasib	ongoing	[92]
RAMP204 NCT05375994	I/II	Avutometinib [VS-6766] (RAF–MEK Inhibitor)	adagrasib	ongoing	[92]
NCT04418167	I	JSI-1187-01 (ERK Inhibitor)	monotherapy/ dabrafenib (BRAF Inh.)	ongoing	
NCT04073680	I/II	serabelisib (PI3K Inhibitor)	canagliflozin (SGLT2 Inh.)	completed	[94,95,96]
NCT04586270	I	TAS0612 (AKT Inhibitor)	monotherapy	ongoing	[94,95,96]
TORCMEK NCT02583542	I/II	AZD2014 (mTOR Inhibitor)	selumetinib (MEK Inh.)	completed	[94,95,96]
NCT01951690	II	defactinib (FAK Inhibitor)	monotherapy	completed	[99]
GALAXY 2 NCT01798485	III	ganetespib (HSP90 Inhibitor)	docetaxel vs. docetaxel alone	terminated (futility)	[106]
NCT01892046	I	SNX-5422 (HSP90 Inhibitor)	chemotherapy	completed	[107]

**Table 3 cancers-16-01447-t003:** Novel treatment strategies.

Trial	Phase	KRAS Inhibitor	Combinations	Status/ Outcomes	References
NCT05737706	I/II	MRTX1133 (KRAS-G12D Inh.)	monotherapy	ongoing	[141]
V941-001 NCT03948763	I	V941 (mRNA cancer vaccine)	monotherapy/pembrolizumab	completed	[144]
NCT04735068	II	hydroxychloroquine (autophagy inhibitor)	binimetinib (MEK Inh.)	ongoing	[145]
CANOPY-N NCT03968419	II	canakinumab (anti-IL-1β agent)	monotherapy/pembrolizumab	terminated (low enrollment)	[149,150,151]
CHORUS NCT04905316	II	canakinumab (anti-IL-1β agent)	durvalumab, chemoradiation	ongoing	[149,150,151]
CANOPY-2 NCT03626545	III	canakinumab (anti-IL-1β agent)	docetaxel vs. placebo + docetaxel	terminated (lack of efficacy)	[149,150,151]
NCT04691817	I/II	tecilizumab (IL-6 Inhibitor)	atezolizumab	ongoing	[149,150,151]
NCT04940299	II	tecilizumab (IL-6 Inhibitor)	nivolumab—ipilimumab	ongoing	[149,150,151]

**Table 4 cancers-16-01447-t004:** Indirect actionable pathways and targets against KRAS-mutated NSCLC.

Pathway	Target	Inhibitor	References
Upstream signaling pathway	EGFR	erlotinib	[61]
	SHP2	RMC-4630 TNO-155	[87]
	SOS 1	BI 1701963	[87]
Downstream signaling pathway MAPK pathway (RAF–MEK–ERK)	RAF	belvarafenib	[88,89,90,91]
	MEK RAF–MEK	binimetinib cobimetinib VS-6766	[88,89,90,91] [92]
	ERK	JSI-1187-01	
Downstream signaling pathway PI3K–AKT–mTOR pathway	PI3K	BKM120 serabelisib XL147	[94,95,96]
	AKT	TAS0612	[94,95,96]
	mTOR	AZD2014	[94,95,96]
Other pathways	FAK	defactinib	[99]
	HSP90	ganetespib SNX-5422	[105,106] [107]
	CDK 4/6	palbociclib	[74,108,109,110]
	DDR1		
	YB-1—RSK	LJI308	
Immune system	ICIs	nivolumab atezolizumab	[113] [115]
Metabolic rewiring	autophagy	hydroxychloroquine	[145]
	FASN	TVB-2640	[146]
Tumor microenvironment (TME)	IL-1β IL-6	canakinumab tecilizumab	[148,149,150,151]

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
