# Peer review of "Emerging Therapies in Kirsten Rat Sarcoma Virus (+) Non-Small-Cell Lung Cancer"

_cancers, 2024, doi:10.3390/cancers16081447_

Round 1

Reviewer 1 Report

Comments and Suggestions for Authors

Dear Authors, congratulations for taking a very important topic of KRAS targeting in cancers, including NSCLC is of high significance: and documentig various ongoing clinical trials. However, here are few places where some additional information needs to be incorporated:

1. In indirect or alternative actionable targets against KRAS,

a. DDR1

DDR1 is a very important and new emerging target. Please include this with various significant citations: Here are a few:

Altaf R, Ilyas U, Ma A, Shi M. Identification and validation of differentially expressed genes for targeted therapy in NSCLC using integrated bioinformatics analysis. Front Oncol. 2023 May 31;13:1206768. doi: 10.3389/fonc.2023.1206768. PMID: 37324026; PMCID: PMC10264625.

Gupta K, Jones JC, Farias VA, Mackeyev Y, Singh PK, Quiñones-Hinojosa A, Krishnan S. Identification of Synergistic Drug Combinations to Target KRAS-Driven Chemoradioresistant Cancers Utilizing Tumoroid Models of Colorectal Adenocarcinoma and Recurrent Glioblastoma. Front Oncol. 2022 May 18;12:840241. doi: 10.3389/fonc.2022.840241. PMID: 35664781; PMCID: PMC9158132.

Yang SH, Baek HA, Lee HJ, Park HS, Jang KY, Kang MJ, Lee DG, Lee YC, Moon WS, Chung MJ. Discoidin domain receptor 1 is associated with poor prognosis of non-small cell lung carcinomas. Oncol Rep. 2010 Aug;24(2):311-9. doi: 10.3892/or_00000861. PMID: 20596615.

b. YB-1

Khozooei S, Veerappan S, Toulany M. YB-1 activating cascades as potential targets in KRAS-mutated tumors. Strahlenther Onkol. 2023 Dec;199(12):1110-1127. doi: 10.1007/s00066-023-02092-8. Epub 2023 Jun 2. PMID: 37268766.

c. Other upstream and downstream targets:

Cucurull M, Notario L, Sanchez-Cespedes M, Hierro C, Estival A, Carcereny E and Saigí M (2022) Targeting KRAS in Lung Cancer Beyond KRAS G12C Inhibitors: The Immune Regulatory Role of KRAS and Novel Therapeutic Strategies. Front. Oncol. 11:793121. doi: 10.3389/fonc.2021.793121

d. miRNA and KRAS

e. Metabolism and KRAS

2. In the section of KRAS biology, it would be good to briefly discuss co-association of WNT and KRAS pathways, how they co-modulate each other and how they can be co-targeted. Please cite following article, and few back-references from it on how KRAS and WNT corporate with each other to cause tumor stemness and treatment resistance.

Gupta K, Jones JC, Farias VA, Mackeyev Y, Singh PK, Quiñones-Hinojosa A, Krishnan S. Identification of Synergistic Drug Combinations to Target KRAS-Driven Chemoradioresistant Cancers Utilizing Tumoroid Models of Colorectal Adenocarcinoma and Recurrent Glioblastoma. Front Oncol. 2022 May 18;12:840241. doi: 10.3389/fonc.2022.840241. PMID: 35664781; PMCID: PMC9158132.

3. It would be good to make a comprehensive table including: Indirect actionable pathways (ECM, immune, metabolism etc) and, receptors (RTKs) or proteins (EGFR, DDR1, SOS, SHC2, PIK3, mTOR, RAF, etc.), that can be targeted against KRAS-driven (KRAS mutant) cancers, with examples of some inhibitors against these, with citations.

Reviewer 2 Report

Comments and Suggestions for Authors

Dear Editor and Authors,

It was my pleasure to review this review manuscript titled “Emerging therapies in KRAS (+) NSCLC” by Dr. Anastasia Karachaliou and Professor Syrigos’ team from “Sotiria” Chest Diseases Hospital in Athens, Greece.

In this extensive review article the authors present an overview of novel KRAS targeted oncotherapies in NSCLC. They very thoroughly present the mechanics of action of the KRAS pathway, the encountered mutations and the different therapies currently approved or under investigation.

The manuscript is well structured and written up. The language is clean and understandable.

I only have one question for the authors:

Do they feel that KRAS oncogene expression can affect the response of patients to traditional platinum based chemotherapy and or immunotherapy and if yes, what is the mechanics - pathway targets supposedly involved in this? It would be of interest to see if there is a difference in response to traditional/already established treatments in NSCLC patients with KRAS expression and without! Maybe this can be added as an additional section in the work.

Apart from this minor personal query, I am quite satisfied by the work and would be happy to recommend its’ publication. Kind regards to all.   

Comments on the Quality of English Language

Language is quite good. Minor edits if any.

Reviewer 3 Report

Comments and Suggestions for Authors

This is a well done review about KRAS and emerging therapies for patients with NSCLC. The authors made an extensive review on the topic, covering the biology of KRAS, mutations, co-mutations, in addition to a critical analysis of the resistance mechanisms of KRAS G12C inhibitors. Table summarizes the main agents and corresponding clinical trials. The text is written with scientific rigor, in a clear, coherent manner, facilitating reading and interest for readers in general.

Comments on the Quality of English Language

The English language is fine.

Reviewer 4 Report

Comments and Suggestions for Authors

In this interesting manuscript, authors review present current and emerging therapeutic approaches for targeting KRAS mutation subtypes. The manuscript is well structured and written providing information about KRAS biology, direct targeting of KRAS, resistance to KRAS G12C inhibitors, indirect inhibition of KRAS, immunotherapy in KRAS mutated NSCLC, KRAS G12V and G12D mutations, as well as other emerging therapies. I only have 2 minor comments to improve the manuscript prior to be considered for publication in cancers:

1) Authors should improve the quality of figure 1

2) Authors should include references in tables
